# Analogue Meets Digital: History and Present IT Augmentation of Europe's Largest Landscape Relief Model in Villach, Austria

**Manfred F. Buchroithner** 

Institute for Cartography, Dresden University of Technology, 01069 Dresden, Germany;
manfred.buchroithner@tu-dresden.de

**Abstract:** Brought to completion in 1913 after a production time of 24 years, the landscape relief model of Carinthia (Kärnten), on display in Villach, Austria, is, at 182 m$^2$, the largest of its kind in Europe. It is painted with nature-like land-cover information and presents the whole federal state of Carinthia and its surroundings including Austria's highest peak, Großglockner, at a scale of 1:10,000. From 2016 to 2018, a series of computer-generated and partly computer-animated educational contents for rental tablets as well as for projection onto the terrain model and above it have been produced. Their topics are briefly presented. The described Relief von Kärnten is also a paramount example and master copy of how to improve the attractivity of historical physical landscape relief models by means of state-of-the-art information technology. The article is, furthermore, meant to raise awareness for a piece of "geo-art", which is worth being known at an international scale by both experts and laymen.

**Keywords:** landscape relief model; true-3d; largest terrain model of Europe; history of landscape relief models; 3d terrain modelling and representation; digitally animated projection upon landscape relief model; physical landscape relief models and its augmentation

---

## 1. Introduction

It may be considered a strange fact that the largest landscape relief model of its kind in Europe is—even in times of a revival of 3D visualizations—almost unknown amongst professionals, not to mention the laymen. This is all the more surprising as this piece of educational "geo-art" is located in the center of Villach, a city in the southern Austrian Alps that is probably *the* most important traffic hub there with rail- and highway connections to Germany, the Region of Vienna, Hungary, the Balkans, and Northern Italy. Thus, apart from being an excellent means of geographic education (and beyond), it is also a tourism highlight, an attraction for sightseeing par excellence. Note that the above formulation "relief model of its kind" refers to the detailed scale of 1:10,000, the modelling of the alpine relief which represents a particular challenge to its producers, the meticulous painting and—last but not least—the consideration of the earth curvature.

The present paper tries, for the first time, to describe this three-dimensional cartographic monument at a scientifically sound level and explains the history of origins as well as peculiarities of the Relief von Kärnten. Although the author has not been involved in the described IT-based augmentation with up-to-date digital information, he was, in a collaborative manner, encouraged by staff of the Municipal Museum and Archive of the City of Villach and the CEO of *edufilm und medien Inc.* to make the facts described in the present article known to both experts and the public. The majority of history therein have been retrieved from an article authored by Matthias di Gaspero, published in 1931 in a book by Erwin Stein (Ed.) with the title, Die Städte Deutschösterreichs. Eine Sammlung von Darstellungen der deutschösterreichischen Städte und ihrer Arbeit in Wirtschaft, Finanzwesen,

Hygiene, Sozialpolitik und Technik. Band VI: Villach". Valuable information stems from interesting personal and remote discussions with staff of the Municipal Museum and Archive of the City of Villach and the CEO of *edufilm und medien Inc.* This applies to all statements where no explicit literature references are given in the text.

## 2. Analogue Relief

### 2.1. Initiation

The initial idea to generate a detailed, topographically exact terrain model of the Austrian federal state of Carinthia (German: *Kärnten*) and its surroundings was raised around the year 1885. In 1889, the Director of the former Imperial Technical College (German: *K. u. K. Fachschule*) in Villach, *Ernst Pliwa* (22 August 1857–12 September 1928) proposed a motion during the Annual General Assembly of the Section Villach of the German and Austrian Alpine Club (*Deutscher und Österreichischer Alpenverein*) for the creation of this terrain model. At the same assembly, this idea was accepted. Subsequently, the City of Villach held out the prospect of a building site. At the very beginning, it was intended that the whole geosculpture should have an extension of approximately $10 \times 20$ m, thus basing it on a horizontal scale of 1:7500, and a vertical exaggeration factor of 3.41. Consequently, the peak of Großglockner, Austria's highest mountain, was supposed to have a height of 1.8 m. These plans were, however, later changed (cf. text below). It was then a group of some teachers of the College, its caretaker, and several students who eventually, on 1 April 1891, began with the actual production on the proposed municipal ground. The material expenses were supposed to be covered by voluntary donations. By the year 1891, numerous donations had already reached the respectable sum of more than 3000 guilders (German: *Gulden;* [1]). At that time, this initiative was really something extremely rare, as a comparative view at the famous Musée des Plans-Reliefs in the Hôtel National des Invalides in Paris (www.museedesplansreliefs.culture.fr) shows. Similar to the "living model" of a karst lake at the Museum in Cerknica, Slovenia, (https://jezerski-hram.si/en/museum_of_lake_cerknica/living_model_of_cerknica_lake/), this landscape model also intended to give a basic understanding of geomorphology, an intention that has, in recent years through digital augmentation, still been boosted.

*Ernst Pliwa*, the proposer and initiator of the landscape relief model, was appointed Director of the Villach Technical College on 1 August 1884. Under his management, it developed into a model school. He basically designed the Relief von Kärnten, a masterpiece that, according to a contemporary source (1918, see Figure 1), was not only known in Carinthia, but also extending far beyond. For his initiative and active role in the production of the landscape relief model, he was, at its opening ceremony, appointed Honorary Citizen of the City of Villach (Figure 1). Furthermore, Pliwa rendered great services to the restructuring of the Austrian advanced education system (https://www.geschichtewiki.wien.gv.at/Ernst_Pliwa).

**Figure 1.** Entry about Ernst Pliwa into the Book of Honorary Citizens of Villach conducted from 1938 to 1941. For explanation, see text above.

## 2.2. Production of Landscape Relief Model and Construction of Shelter Hall

In 1890, the total presumptive production time was still optimistically estimated to be three to four years at the maximum. After lengthy discussions, eventually a horizontal scale of 1:10,000 and a vertical scale of 1:5000 were chosen, implying a twofold height exaggeration. This vertical super elevation was meant to result in a more realistic accentuation of the pronounced relief in the Alps, which better corresponds to the common "bottom-up" viewing habits, although the slopes of the mountains, thus, do not reflect their true inclination. For the entire depicted region, a size of $19.5 \times 9.35$ m$^2$ was eventually selected, for the sake of easier production by breaking it down into portions of $1.8 \times 1.4$ m$^2$, measures that correspond to the size of the magnified 1:25,000 map sheets [1].

As a cartographic basis for the generation of the physical relief model, the original surveys of the *k. und k. Militärgeographisches Institut* (Imperial Military Geographic Institute in Vienna) at a scale of 1:25,000 were used, which cover Carinthia and its surroundings in 63 map sheets. With the aid of a pantograph, the 100 m contour lines were then transferred onto cardboard at a scale of 1:25,000. Subsequently, these thin layers were in their correct geoposition mounted upon "bridges" of fine-planed wooden bars and discs, then fixed over each other in layers, and eventually, the remaining intermediate spaces were filled with pug, thereby carefully elaborating the characteristic relief features of the terrain [1].

At this point, it may be worthwhile to mention that for a scale of 1:10,000 nominally, each 10 m elevation contour—the usual equidistance at scales of 1:10,000 (cf. [2])—would, with an exaggeration factor of 2, have corresponded to a plank thickness of 2 mm. These "slices" were subsequently supposed to be cut out from panels.

From all the individual wood-clay models, casting molds out of gypsum were made, which were then grouted with white cement. Thanks to a significant financial support by the Savings Bank of Villach (Villacher Zeitung, 4.9.1910, p. 5) in autumn 1892, the basement construction was completed, and under the guidance of College *Fachlehrer* (German for: specialist subject teacher) *Dominik Haubner,* the assembling and mounting of all the *Reliefsektionen* began. A remarkable fact is that for this work, according exactly to the four cardinal directions, the curvature of the earth was taken into consideration. This amounted in a maximum height difference of 35 mm in cross direction and 70 mm lengthways [1]. The basement was a concrete foundation especially prepared for the landscape relief model in the park beside the college, the *Schillerpark* in Villach downtown.

During these works, a provisional low-level roof was supposed to protect the growing landscape relief model from rain and snow. It was, however, anything but waterproof and, thus, the condition of the relief model deteriorated repeatedly. At an early stage the construction of an "*ordentlicher Pavillon*", an appropriate shelter building was envisaged, the costs of which were estimated to amount to approximately 30,000 Crowns. Its size was planned to be 12.5 m × 25 m. Furthermore, rolling shutters on every side were envisaged in order to compensate for the changing elevation and azimuth of the sun illumination. Regarding the vertical viewing angle of the visitors, the floor of the corridor around the landscape relief model was planned in a way that, assuming an average tallness, an eye height of 2.5 m was permitted. In addition, plans were made for a catwalk gallery at a height of 5 to 6 m ([3], p. 5; cf. also Figure 2 and text further below).

After the change of Director Ernst Pliwa from the Villach Technical College into the Ministry of Education in Vienna at the beginning of 1899, the foreman of the college, *Josef Paikert,* assumed the guidance of all activities, and after his untimely death, the very capable school caretaker *Josef Rautter* took over, who had already been responsible for the production of the casting molds and the molding of the individual sections since the beginning of the work. His son, a graduate of the Technical College in Villach, assisted him, and thus, in 1907, the mounting of all sections of the landscape relief model covering Carinthia proper could be completed. In order to backfill the regions outside Carinthia, *Rautter*, his son and the sculptor *Peter Piron*, also an alumnus of the Villach Technical College, completed the whole 182 m$^2$ area based on map sheets provided by the Chairman of the Section Villach of the Austro-German Alpine Club, *Josef Aichinger*.

Since the Section Villach of the German and Austrian Alpine Club could not procure the funds for the completion and the shelter-building construction of the landscape relief model, in the same year of 1907, the *Relief von Kärnten* was committed to the City of Villach. Subject to the influence of *Hofrat* (an Austrian title for Privy Councillor) *Ernst Pliwa*—then *Sektionschef* (an Austrian title for head of ministerial department) at the Ministry of Public Works in Vienna—and *Feldmarschalleutnant* (German for Field Marshal Lieutenant) *Othmar Crusiz*, substantial donations from this ministry allowed the stylish and functional construction of the shelter building. It was erected according to the plans of the Municipal *Baurat* (an Austrian title corresponding to Chief Construction Engineer) of Villach *Rudolf Müller* and finished in 1912, i.e., 19 or 20 years later than initially assumed [1]. The total expenses eventually amounted to 20,000 Crowns.

From the above statements, it can already be inferred that the whole production process proved to be more intricate and complex and, thus, much more time-consuming than expected. The major reason for this was that it turned out to be extremely difficult to obtain sufficiently detailed and accurate maps at the required scale, in particular for the Slovenian and Italian regions [4]. Interestingly enough, for the latter ones, it was almost impossible to acquire adequate terrain information.

In 1913, the painting of the relief model became due. After short negotiations, the internationally renowned geosculptor *Paul Gabriel Oberlercher* (6 January 1859–11 February 1915) was persuaded to accomplish this task [4]. He worked devotedly from September 1912 until June 1913, except for the severest winter months. *Oberlercher*, born as the son of a country teacher and sacristan in Sankt Peter im Holz not too far from Villach on 1 January 1859, was an Austrian elementary school teacher. He worked and lived in Carinthia and died only one year after finishing the landscape relief model of Carinthia

on 11 February 1915 in its capital Klagenfurt. Although he learned the art of terrain modelling as an autodidact, he may be considered the best Austrian modeler (although the world-famous geosculptor *Toni Mair*, 30 April 1940–8 August 2015, who lived in Switzerland, was actually also an Austrian citizen, born near Meran but grown up in Innsbruck). He used the same techniques as the famous Swiss relief-model artists of that time to create a total of 37 landscape relief models. Apart from in-depth field surveys resulting in numerous landscape sketches, he also used a self-constructed theodolite. His relief model of the Ankogel-Hochalm Massif (1889), where the *Oberlercherspitze* (3051 m) was named after him, brought him international recognition. In 1894, he finished his main work, the large-size model (7.0 m × 3.5 m) of the Großglockner, Austria's highest peak, at a scale of 1:2000. In the following years, he created numerous landscape relief models of regions in and outside Europe, frequently only based on the information of moderately reliable maps and few pictures. Many of them were then copied for universities and museums. More than 100 copies of his *Schulrelief von Kärnten* (Carinthia) (1893/94) went into the schools of Carinthia (cf. also [5,6]).

*Oberlercher* emphasized the rocks in relation to the surrounding terrain by using distinct brownish-grey colors, and glaciers had the most natural appearance. Also, individual rocky ledges (German: *Schrofen*), talus slopes, and high-altitude alpine pastures got their specific, natural colors. The coniferous forests of the Alps were painted in a darker green, river forests in the valleys and fields got lighter green hues, and swamps a brownish green color. Running waters were drawn in dark, standing waters colored in light blue, a pinkish grey color was used for settlements, and even isolated farmsteads of alpine farmers were precisely indicated in order to give an impression of the altitude of the productive soil. In contrast to contemporary maps, railways were not painted in black but in the most intense of all colors, in red, whereas roads and paths were, for identical reasons, indicated in black.

The artistic and realistic painting, which was completed in laborious manual labor, allowed us to experience the landscape relief model from the surrounding floor as a sort of micro-scenery and from the gallery above as a sort of map projection (cf. [1]).

The actual painting was limited to the 1913 extent of Carinthia including the border seam to East Tyrol (German: *Osttirol*) in order to allow a complete coherent surface representation of the Großglockner Massif. The neighboring Habsburg countries of Salzburg, Styria (German: *Steiermark*), Slovenian Kranj (German: *Krain*), as well as Italy were left white. The former Habsburgian Crown Land Carinthia (*Kärnten*) also comprehended the *Mieß* (Slovenian: Meza) Valley, the region of *Seeland* (Slovenian: Jezersko), and the *Kanaltal* (Italian: Val Canale) down to *Pontafel* (Italian: Pontebba). After World War I, these regions were assigned to Yugoslavia and Italy. This is why, today, the  landscape relief model shows a rather wide border seam in the South. In 1918/1919, the regions conveyed away were whitewashed. When in World War II Upper Kranj (German: *Oberkrain*), the part of Slovenia annexed by the *Deutsches Reich* was, for a short time, supposed to be assigned to Gau Carinthia (German: *Gau Kärnten*, the Nazi Germany administrative division), the idea came up to extend the landscape relief model correspondingly. Fortunately, this has been postponed by the municipal administration of Villach until "*nach dem Endsieg*" (after the *Ultimate Victory*), so that today's situation remained [4].

In order to obtain a most natural (over)view of the landscape, a catwalk or gallery at a height of about 3 m was built. From its elevated positions, the vertical exaggeration recedes, and the impression of the landscape relief becomes sort of normal, like from a prominent peak. Further, the size and the elevated viewing possibility allow for both an excellent overview as well as a quick orientation, and the distortion due to the height exaggeration vanishes. With its final size of 182 m$^2$, the landscape relief model of Carinthia was, and still is, the largest of its kind in Europe. Today, both the sheltering structure and the relief model are protected monuments.

The "*Reliefgebäude*" or "*Pavillon*" (shelter building) was constructed in 1912, based on the plans of the Villach *City Baurat* (an Austrian title for municipal construction engineers), the civil engineer *Rudolf Müller*. Initially, it received natural light through a large glass roof (Figure 2a–c). Since this sort of natural illumination did not prove itself in practice due to interaction with the sun radiation, only a few years later, the building got a roof membrane made of tin and an electric lighting (Figure 3) [4].

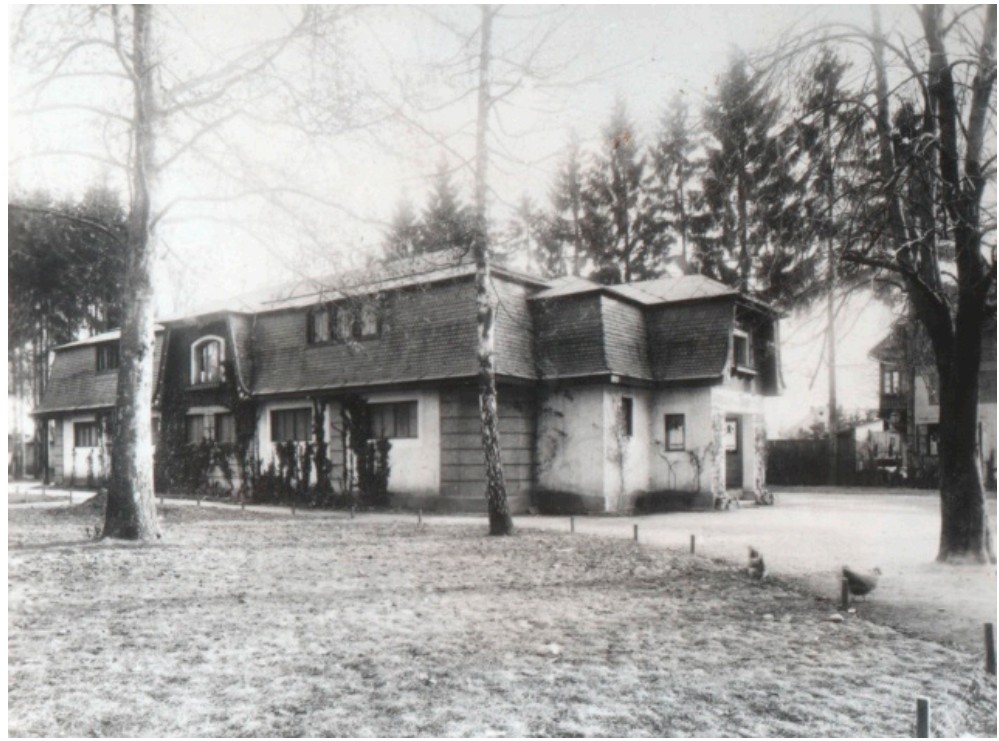

(**a**)

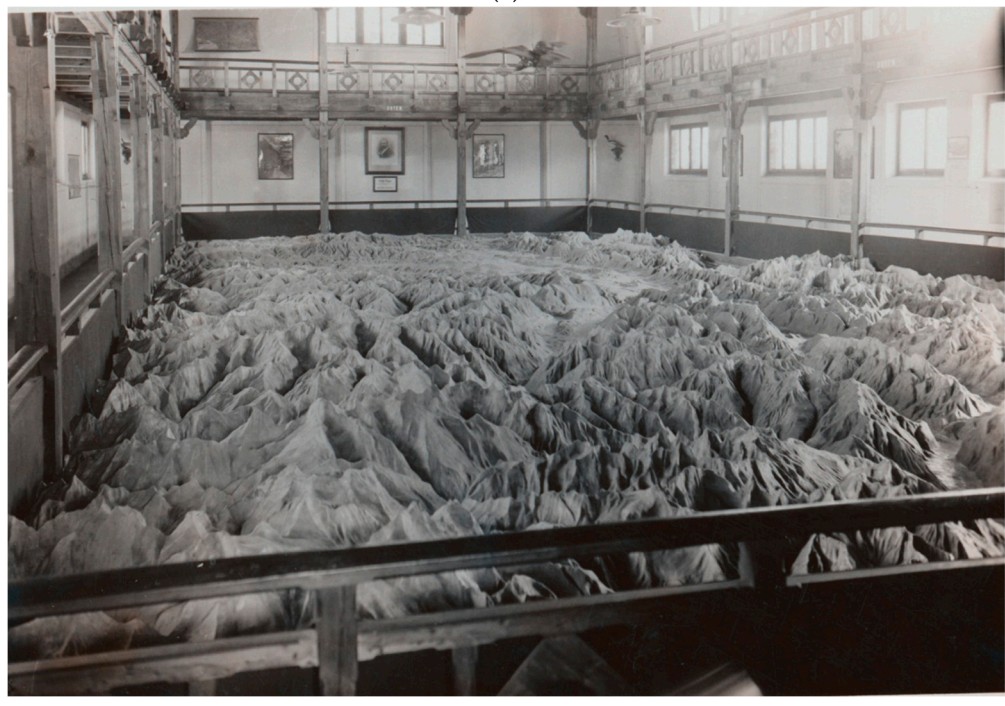

(**b**)

**Figure 2.** *Cont.*

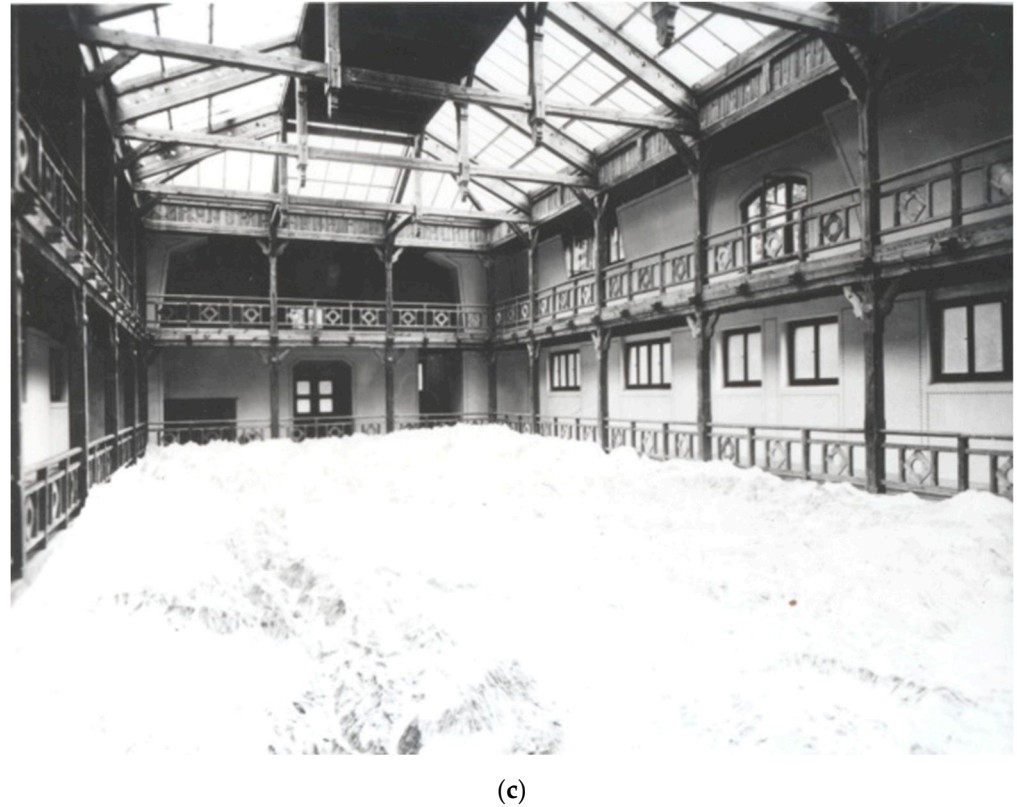

(**c**)

**Figure 2.** The initial shelter building for the *Landscape Relief Model of Carinthia* with glass roof around 1925. (**a**) Exterior view and (**b**,**c**) interior views. In (**b**) and (**c**), the catwalk gallery can be seen. (**c**) also visualizes the overly bright illumination of the landscape relief model through the glass roof at noon. Photographs: Municipal Museum and Archive of the City of Villach.

Three weeks after the official opening of *the Relief von Kärnten,* more than 1000 visitors were counted [7] (p. 5). Less than only four weeks after the opening, the landscape terrain model had been visited by more than 2000 persons. According to the local newspaper *Villacher Zeitung* of 24 August 1913, all visitors were amazed and delighted by the splendidness of this creation. The author of the article praised the natural appearance and uniqueness of the object and underlined its attractiveness for the local population as well as for tourists. It was predicted that the *Relief von Kärnten* would, in the future, act as a prominent tourist attraction. The *volkswirtschaftlicher Verein* (national-economic association) "*Oberkärnten*"("*Upper*" = Western = more alpine "*Carinthia*") made it their business to promote this "sculptural piece of art" in Austrian newspapers and newspapers abroad in order to draw attention to this unique object of interest ([8], p. 5).

In 1986/1987, the building got a thorough renovation and the landscape relief model was also refurbished. Two years later, in 1988/1989, the already yellowed terrain painting was restored by the Tyrolian geosculptor *Jörg Covi*. Within the scope of this renovation, the previously white neighboring regions of Carinthia also got a topographically correct coloring, so that today the whole area of 182 m$^2$ shows a "real landscape" and not only the whitish terrain relief [4].

The large-area *Relief von Kärnten* owes its existence essentially to an alpinism, educational, and local-history motivation but—to be frank—also, to quite some extent, a patriotic–nationalistic motivation. At times when eidetic cartographic (3D) visualizations did not yet exist, not to mention aerial and satellite imagery or virtual digital panoramas—things that can be taken for granted today, this *Landscape Relief Model of Carinthia* played a seminal role (Figure 4, cf. [4]). Also today, during a period of ever increasing digitalization, this genuine landscape relief model, following the postulation of the world-famous geosculptor Toni Mair, has to be considered both a piece of excellent *handicraft and*

*a scientific work of art* (cf. [9]), even if we are not taking into account the magnificent value-adding by means of most advanced IT technology (see next chapter).

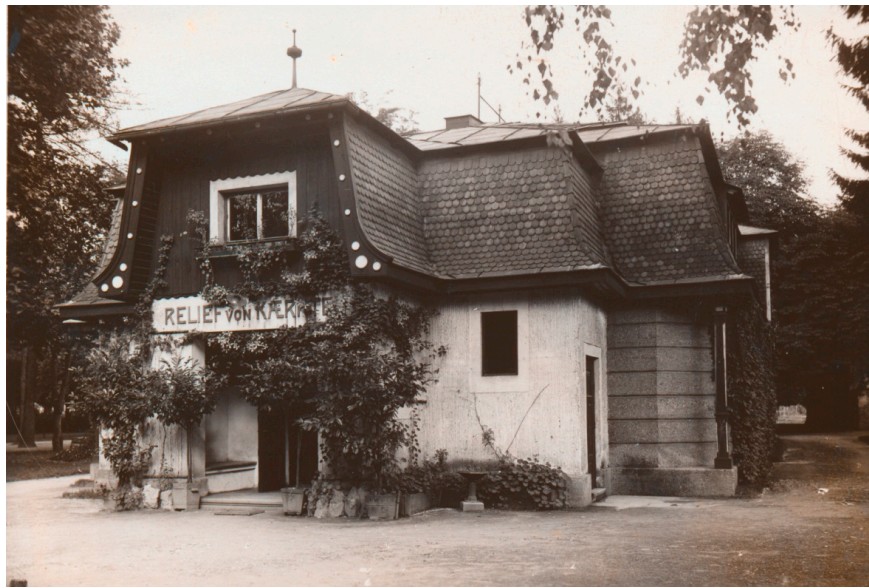

**Figure 3.** Shelter building for the *Landscape Relief Model of Carinthia* with changed roof membrane made of tin (around 1925). Photograph: Municipal Museum and Archive of the City of Villach.

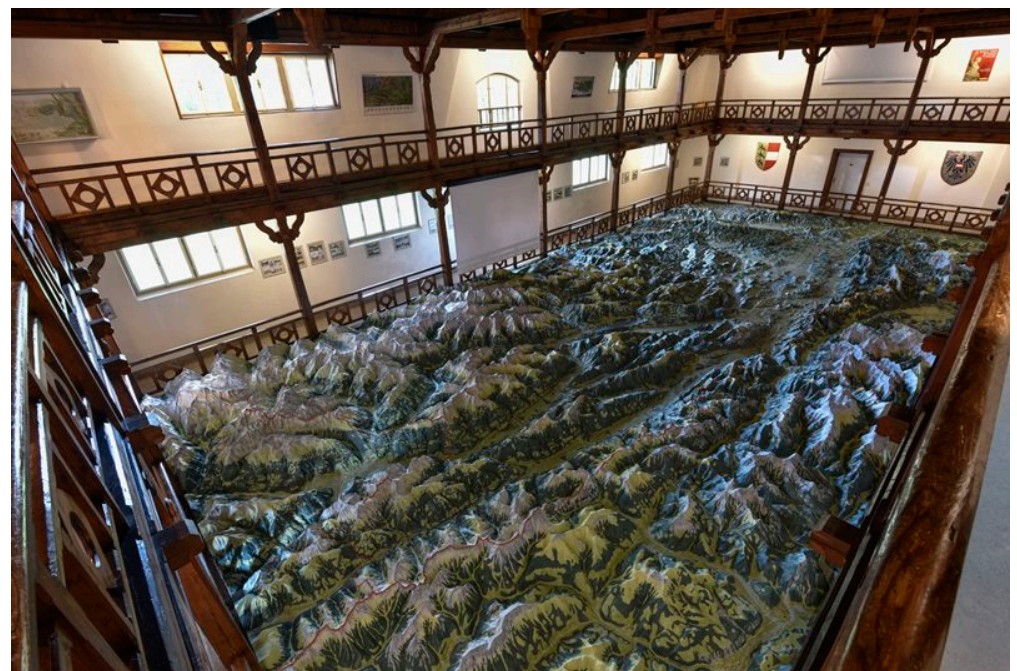

**Figure 4.** Present-day impression of *Relief von Kärnten* without IT augmentation. In the upper center, the vertical screen for video projection can be seen. From the Internet Presentation of the Municipal Museum and Archive of the City of Villach.

## 3. The Digitally Augmented Relief

*Digital Augmentation of the Physical Relief Model*

Both the landscape relief model and its shelter building are *protected monuments*. Nevertheless, in the years after 2010, a known Carinthian landscape photographer came up with the idea of

"augmenting" the relief model with some of his landscape pictures. Although this was, for protective reasons, not possible, the idea was born to "value-enhance" the landscape relief model by adding complementary geo-information. So, for some years, the Municipal Museum of Villach began to investigate technical possibilities for materializing an improved modern form of museal communication that allows an up-to-date and innovative presentation and, at the same time, acknowledges the statutory provisions of legal monumental protection (Figure 5).

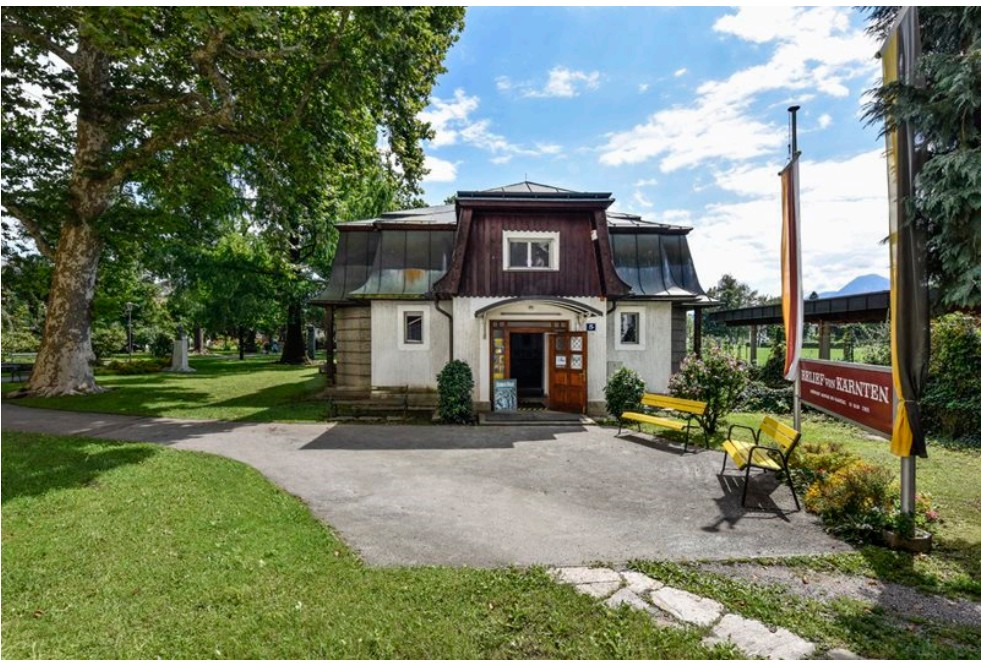

**Figure 5.** Present-day view of the *Relief von Kärnten Pavillon* at the Schillerpark in Villach. From the Internet Presentation of the Municipal Museum and Archive of the City of Villach.

Thus, in 2015, the museum approached the multimedia and film production company *edufilm und medien GmbH* in Bleiberg-Kreuth near Villach to develop a conception that, on the one hand, implements modern information and presentation technology, and on the other hand, takes care of the special requirements and challenges of monumental protection. In cooperation with the internationally active company for media and communication technology *PKE Electronic AG* in Vienna/Klagenfurt, a total of seven high-performance projectors were mounted above the relief model. Additionally, a server and a sound system take care of the content that has to be displayed onto the landscape relief model. Six projectors (two rows of three) cover the 182 m$^2$ of the model, another projector displays information onto a newly installed vertical screen above the geosculpture. The idea to project information onto the relief called for blackout blinds in order to allow high-quality presentations. The *m3* software created by *multi-media-machines* in Rennweg am Katschberg, Carinthia, is responsible to distribute the content correctly to the installed hardware (Table 1).

After the installation of the required hard- and software, from 2016 to 2018, *edufilm und medien GmbH* developed a series of educational computer-animated contents for the projection mapping. As it had already turned out during the production of the projection of the historical development of the City of Dresden, Germany, onto a terrain model of this region by M. F. Buchroithner and his team in 2006 (cf. www.stadtmuseum-dresden.de/userfiles/relief.swf), also in Villach, *the* biggest challenge was the precisely accurate pointing at the relief. The objective was to show the visitors particular locations, settlements, mountain peaks, the border of Carinthia, individual districts, and more. The generation of the pixel-precise digital presentation models that allowed the highly accurate positioning of projected objects on the relief model was achieved through painstaking manual labor. As a result, it is now

possible to show exactly geocoded additional thematic information on the uneven surface of the model by means of high-quality projectors.

**Table 1.** Technical documentation of the IT part of the landscape relief model of Carinthia.

| Number | Devices |
|:------:|:-------:|
| 6 | *NEC PX800X projectors* for the landscape relief model |
| 6 | special wide-angle optics for the coverage of the whole relief model *NP31ZL Lens* |
| 1 | *NEC PA722x* projection screen, linked with |
| 1 | optical projector *NP13ZL Lens* |
| 1 | *Integris System 4U / 55cm, 16GB,* 2x *Matrox C680 Graphic Card* in the server for the rendering of the film contents for the projectors |
| 1 | *m3 Software* for the organization of the media contents |
| 1 | *Soundcraft UI 16 Crossplattform Sound System* |

Due to the fact that today, common HD and 4K products tend to be state-of-the-art in edutainment, the production of geocoded information over an area of 182 m² represented a real challenge to the developers: Tiniest details had to be accurately visualized for both monitors and projectors. Pretty soon, however, it became apparent that with the usual approaches, the visitors could not visually perceive the details due to the size of the relief model. Precisely accurate pinpointing of peaks, for example, is fruitless at scales like the one of the *Relief von Kärnten.* This led to the development of large—sometimes successively shrinking—animations in order to steer the viewing direction of the visitors.

In 2017 and 2018, some special contents were developed, optimized for the newly installed technology. Actually, for each content, two video files supplementing each other are necessary. Therefore, two individual films had to be generated. They are shown frame-precise and simultaneously: The vertical screen above the landscape relief model makes use of the film contents, which are projected complementary to the geocoded computer animations presented on the model. Since September 2016, spectacular sound and light effects provide a new, exciting digital and interactive 3D experience. Thus, the centennial tradition is combined with the most modern high tech. Valleys and rivers, settlements and the road network, and natural phenomena are now made conspicuous via animated projections (Figure 6). Also, 128 top tourist attractions all over Carinthia can be rendered visible through brief glances at a loan tablet.

Between 2016 and 2018, the following educational contents for the landscape relief model were generated (most of this information has been taken from [10]):

- *Kärnten Multimedial* (Multimedia Tour of Carinthia: 8:10 min.) in German *and* English: This multimedia tour presents Austria's southernmost federal state. 3D techniques and special effects help the viewers to discover Carinthia's geography. With the help of projectors, it is easy and impressive to spot the cities, villages, lakes, peaks, and valleys of Carinthia.
- *Das Relief von Kärnten—Das größte Landschaftsmodell Europas* (History of the relief model: 4:20 min.): This short film portrays the origin and making of the physical landscape relief model from its beginning until 1913. Precise information about the measurements, horizontal, and vertical scales are given. Also, *Ernst Pliwa* has a say about the generation of the landscape relief model.
- *Die Reformation in Kärnten* (Reformation in Carinthia: 6:00 min.): How the protestant movement of the 16th century developed in Carinthia.
- *Verkehr in Kärnten* (Traffic in Carinthia: 6:00 min.): The major traffic lines in an alpine high-relief terrain. In ancient times, Villach was already an important communications junction. The film illustrates the development from the Roman road network until today's modern traffic system.
- *Sagen in Kärnten* (Sagas in Carinthia: approx. 10 min.): Project of students of CHS Villach (Centre of Human-Profession Schools in Villach) under supervision by its teachers and *edufilm und medien Inc.*

- *In die Berg bin i gern* (*In den Bergen bin ich gerne*; I love to be in the mountains (an Austrian mountain song title: approx. 6 min.): Again, a film about the alpine world of Carinthia made by students of CHS Villach (Centre of Human-Profession Schools in Villach) under supervision by its teachers and edufilm und medien Inc. In an entertaining way, the lifestyle of the local population and the geolocations of the individual mountain ranges and massifs are presented.
- *Faszination Geologie* (Fascination Geology: 8:45 min.): The mountains of Carinthia are particularly rich in minerals. This educational short film provides an insight into the fascinating world of its subsurface treasures. The (pretty unknown) volcanic region of Carinthia as well as the last Ice Age, its wildlife, and the orogenesis of the Alps and their glaciers are visualized.

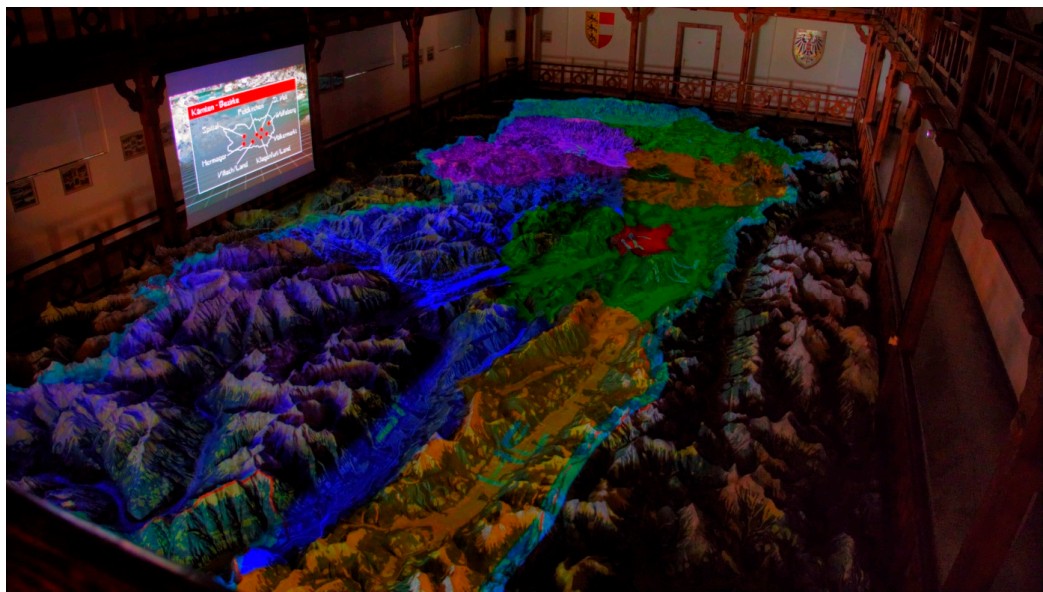

**Figure 6.** Example of simultaneous projections onto the landscape relief model and the screen above the *Relief* giving information about the administrative structure of Carinthia. From the Internet Presentation of the Municipal Museum and Archive of the City of Villach.

Three times a day, between 10:00 and 15:40, these six films can be viewed, with a film every 20 min. At the URL link http://www.villach.at/stadt-erleben/museum-der-stadt/relief-von-kaernten, a glimpse of the computer-animated films offered to the visitors can be obtained.

In addition, objects of interest are explained and described on the visitors' mobile phones in real time, without the need for a QR code or typing in search machines. This is possible because of an advanced technology that, via the camera image of the mobile phone, calculates the visitor's position in space accurately to the mm. This allows the insertion of information about various geo-objects, tourist attractions, and much more, directly into the camera image [2,6].

As a valuable side effect, *edufilm und medien Inc.* provides their gained experience and expertise to school classes. Within educational projects, the installation is presently also used to work on and acquire skills in multimedia projects and, thus, to provide an insight into this innovative technology.

## 4. Discussion, Conclusions, and Recommendations

In the present article, for the first time, the digitally augmented *Landscape Relief Model of Carinthia* has been described. It represents the first case of a relief model of this size with "true-3D" cartographic information. So far, the number of visitors who want to actively interact with the information hub of the installation has not yet caused serious problems. Internal statistics provided by the administration of the Municipal Museum of Villach show, however, that in the period from 2016 (last year without IT augmentation) to 2018 (first year with IT augmentation), the number of visitors

increased by 32%. This implies that with an increasing number of IT-prone visitors, issues like the range of interactivity and the maximum number of visitors simultaneously interacting with the existing database will have to be discussed. Further, the question of digital user input to the system needs to be addressed. Another suggestion worth investigating is the link with the Carnic Alps Geopark located nearby in Carinthia in order to provide dynamic true-3D geological information complementary to the facts presented in the visitor center of the geopark (http://www.geopark-karnische-alpen.at/Besucherzentrum.998.0.html). Every year between 2 May and 31 October, with opening hours of Monday through Saturday from 10:00 to 16:30, the Relief von Kärnten represents a really up-to-date geo-experience of a particular kind and during these opening hours, feedback and suggestions can be forwarded to the operators of this landscape relief model.

**Funding:** This research received no external funding.

**Acknowledgments:** Although the idea to compile, for the first time, a comprehensive documentation about the unjustifiably not so much known "digitally augmented" *Landscape Relief Model of Carinthia* stems from the author, this paper could never been written in the present form without the competent and voluntary assistance of Kurt Karpf, Director of the Municipal Museum and Archive of the City of Villach, Sandra Bertel, also Municipal Museum and Archive of the City of Villach, and Erik Dobat, CEO and mastermind of *edufilm und medien* GmbH, Kreuth bei Bleiberg, Carinthia. Their permanent, friendly encouragement and assistance is much appreciated. The author is extremely grateful to them.

**Conflicts of Interest:** The author declares no conflict of interest.

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
