# Peer review of "Analogue Meets Digital: History and Present IT Augmentation of Europe’s Largest Landscape Relief Model in Villach, Austria"

_mti, doi:10.3390/mti3020044_

Round 1
Reviewer 1 Report
Reviewed paper describes 3D landscape model of Carinthia and it’s extending with multimedia content. I think this is a good example of applying a mixed reality when presenting various geospatial data to general public. I have the following comments to this paper.
Major comments:
- What was the role of the author of this paper in developing multimedia content? This should be mentioned somewhere in the text.
- There is no discussion and conclusion. I think this should be part of every scientific paper. For example, I would expect recommendations for other similar projects. Furthermore, it would be possible to discuss the attendance (number of users) or the user feedback on the multimedia content.
Minor comments:
- rows 68 and 84 – Why is the “Book of Honorary Citizens of Villach“ not cited in References section?
- row 277 – Hyperlink does not work.
- rows 346-349 –Why this paragraph is formatted in square brackets? In addition, it looks like advertising.
Author Response
Reviewer 1:
Major comments:
- What was the role of the author of this paper in developing multimedia content? This should be mentioned somewhere in the text.
SEE LINES 34 AND 35 OF THE MANUSCRIPT.
- There is no discussion and conclusion. I think this should be part of every scientific paper. For example, I would expect recommendations for other similar projects. Furthermore, it would be possible to discuss the attendance (number of users) or the user feedback on the multimedia content.
SEE LINES 356 AND FOLLOWING ONES.
Minor comments:
- rows 68 and 84 – Why is the “Book of Honorary Citizens of Villach“ not cited in References section?
I, AS AUTHOR, BELIEVE THAT THE REVIEWER DOES NOT REFER TO ”ROWS” BUT TO THE NUMBERED “LINES” …
THIS “Book of Honorary Citizens of Villach“ IS A SINGLE-COPY BOOK WITH HANDWRITTEN NOTES ABOUT THE ELECTED HONORARY CITIZENS OF THE CITY OF VILLACH HOSTED IN THE CITY HALL OF VILLACH. HENCE IT CANNOT BE MENTIONED AMONG THE REFERENCES.
- row 277 – Hyperlink does not work.
EVIDENTLY THE HYPERLINK HAS VERY RECENTLY BEEN CHANGED: IN FACT, THE ADDRESS
https://de.wikipedia.org/ wiki/Stadtmuseum_Dresden #Ausstellungen
IS NOT OPTIMUM. HENCE, IT HAS TO BE CHANGED INTO:
www.stadtmuseum-dresden.de/userfiles/relief.swf
SEE LINE 282.
- rows 353-355 –Why this paragraph is formatted in square brackets? In addition, it looks like advertising.
THE REVIEWER SEEMS TO REFER TO THE LINES 349 -352: THUS THE SQUARE BRACKETS HAVE BEEN DELETED. NB: THIS NOTE IS NOT MEANT AS ADVERTISING BUT TO COMPLETE THE INFORMATION ABOUT THE USER-FRIENDLINESS OF THIS DIGITAL AUGMENTATION.

Reviewer 2 Report
The article describes the genesis of the physical model of the mountainous region of Carinthia. The article itself is interesting and well written, but lacks scientific innovation and insight, specifically to the community of the journal it was submitted to. It would be an interesting editorial piece for the cartographic community, but not for a journal on Multimodal Technologies and Interactions. Most of the paper discusses the history of the model, only a very small part discusses modern use and the combination with (multi-modal) digital technologies. And even the parts that describe such technologies only provide an overview and not go into much detail. The author does not provide insights into the use cases and learnings for the multimodal and interaction community.
Furthermore, the style of writing suggests, that the author is very excited about the model, a scientific paper could use a little less enthusiastic language:
"to describe this extraordinary monument"
"the internationally renowned multimedia and film production company"
Most of the historic background, which represents most of the paper, is build upon only one source. Which makes this part appear like a summary of the source. The author mentions conversations with experts, but none are named nor are there any quotes from such "interviews".
The author mentions one of his own works, and references it through a wikipedia articles?
In the references are three anonymous references?
I would strongly suggest submitting this to an outlet like the Kartographischen Nachrichten.
Author Response
Reviewer 2
The article describes the genesis of the physical model of the mountainous region of Carinthia. The article itself is interesting and well written, but lacks scientific innovation and insight, specifically to the community of the journal it was submitted to. It would be an interesting editorial piece for the cartographic community, but not for a journal on Multimodal Technologies and Interactions. Most of the paper discusses the history of the model, only a very small part discusses modern use and the combination with (multi-modal) digital technologies. And even the parts that describe such technologies only provide an overview and not go into much detail. The author does not provide insights into the use cases and learnings for the multimodal and interaction community.
REPRESENTING IN ITS KIND A UNIQUE COMBINATION OF PHYSICAL AND DIGITAL 3D CARTOGRAPHY, THE AUTHOR THOUGHT THAT IT IS JUSTIFIED TO DESCRIBE THE UNUSUAL GENETIC HISTORY OF THE LANDSCAPE RELIEF AND THEN COVER ITS DIGITAL AUGMENTATION (SINCE THERE DO, AT AN INTERNATIONAL SCALE, NOT YET EXIST TOO MANY – EXAMPLES LIKE THIS).
APART FROM INTERNAL DOCUMENTATIONS OF EDUFILM CORPORATION THERE EXISTED NOTHING PUBLISHED. NEVERTHELESS, THE ESSENTIAL INFORMATION HAS BEEN COMPILED IN TABLE 1 AND THE ACCOMPANYING TEXT.
THE VERY RECENT PUBLICATION CITED BELOW WAS NOT YET READY AT THE TIME OF MANUSCRIPT SUBMISSION. IT HAS BEEN INCLUDED NOW: SEE LINES 348 AND 370 ff.:
ALAPP – Advanced Limes Applications for Smartphones
ERIK DOBAT and SANDRA WALKSHOFER, edufilm und medien GmbH, Austria
PATRICIA WEEKS and CARSTEN HERMANN, Historic Environment Scotland, Scotland
LYN WILSON and ALASTAIR RAWLINSON, Centre for Digital Documentation and Visualisation, Scotland
CHRISTOF FLÜGEL, Landesstelle für die nichtstaatlichen Museen in Bayern, Deutschland MARKUS GSCHWIND, Bayerisches Landesamt für Denkmalpflege, Deutschland
Published in: CHNT 22, 2017 Publication date: 2019. 8 pages.
Proceedings of the 22nd International Conference on Cultural Heritage and New Technologies 2017.
CHNT 22, 2017 (Vienna 2019). http://www.chnt.at/proceedings-chnt-22/
ISBN 978-3-200-06160-6
Editor/Publisher: Museen der Stadt Wien – Stadtarchäologie
Editorial Team: Wolfgang Börner, Susanne Uhlirz
Furthermore, the styLe of writing suggests, that the author is very excited about the model, a scientific paper could use a little less enthusiastic language:
"to describe this extraordinary monument"
SEE REPALCEMENT ON LINE 32:
"the internationally renowned multimedia and film production company"
THE TERM “internationally renowned” HAS BEEN DELETED.
Most of the historic background, which represents most of the paper, is build upon only one source. Which makes this part appear like a summary of the source.
THE PAPER REPRESENTS FOR THE FIRST TIME A COMPREHENSIVE DESCRIPTION OF THE GENERATION OF THIS LARGEST MOUNTAIN RELIEF MODEL OF EUROPE, DIGITALLY AUGMENTED IN 2018. IN FACT – AS MENTIONED IN THE INTRODUCTION AND LATER – THE INFORMATION GIVEN STEMS FROM A SERIES OF SOURCES, I.A. FROM HISTORICAL LOCAL NEWSPAPERS, WHICH HAVE IN THIS WAY NEVER BEFORE BEEN COMPILED.
The author mentions conversations with experts, but none are named nor are there any quotes from such "interviews".
THIS IS NOT TRUE.
THE AUTHOR LED TALKS WITH THE INDIVIDUALS MENTIONED AT THE END OF THE “INTRODUCTION”. THEIR NAMES ARE ALSO GIVEN UNDER “ACKNOWLEDGEMENTS” IN THE LINES 371 – 377.
The author mentions one of his own works, and references it through a wikipedia articles?
THIS IS NOT TRUE.
THE ONE AND ONLY TIME THE AUTHOR REFERS TO ONE OF HIS PREVIOUS ACTIVITIES IN THE VERY FIELD OF INTERACTIVE 3D CARTOGRAPHY HE REFERS TO A HYPERLINK IN THE OFFICIAL WEBSITE OF THE STADTMUSEUM OF DRESDEN, SINCE THIS IS AN OBJECTIVE SOURCE, AND NOTHING HAS SO FAR BEEN PUBLISHED ABOUT THIS MANY YEAR OLD CONTRACT WORK FOR THE DRESDEN MUNICIPAL MUSEUM. SEE ALSO COMMENT TO REVIEWER 1.
In the references are three anonymous references?
FOR THESE ARTICLES IN HISTORICAL NEWSPAPER EDITIONS NO AUTHORS ARE GIVEN.
I would strongly suggest submitting this to an outlet like the Kartographischen Nachrichten.
DECISION IS UP TO THE EDITORS OF THE ANTICIPATED VOLUME “INTERACTIVE 3D CARTOGRAPHY”.
Reviewer 3 Report
I read this paper with great interest even though I am not an expert on the subject. As I am working on geotourism, I think that it could be interested to know the links with this model and the "Geopark Karnische Alpen". Villach is at the heart of the geopark and there must certainly be interactions between these two structures. It could be interesting to develop this aspect ? And give some informations about the number of people who now visit the Relief Model.
I think the introduction could show more that the relief plans were something important not the past (http://www.museedesplansreliefs.culture.fr) and to understand geomorphological phrnomenon (https://jezerski-hram.si/en/)
There is no conclusion in the paper may be it could be the opportunity to develop something about the geopark which renew local interest and dynamic to an old heritage ?
Author Response
Reviewer 3
I read this paper with great interest even though I am not an expert on the subject. As I am working on geotourism, I think that it could be interested to know the links with this model and the "Geopark Karnische Alpen". Villach is at the heart of the geopark and there must certainly be interactions between these two structures. It could be interesting to develop this aspect ?
SEE LINES 367 FF.
And give some informations about the number of people who now visit the Relief Model.
SEE LINES 363 FF.
I think the introduction could show more that the relief plans were something important not the past (http://www.museedesplansreliefs.culture.fr) and to understand geomorphological phrnomenon (https://jezerski-hram.si/en/)
SEE LINES 61 FF.
There is no conclusion in the paper may be it could be the opportunity to develop something about the geopark which renew local interest and dynamic to an old heritage ?
SEE COMMENTS OF REVIEWER 1 AND LINES 356 OF THE EDITED MANUSCRIPT.
Round 2
Reviewer 1 Report
All my sugestions have been responded in the text of the paper and author commented them in a cover letter.
I recommend accepting the paper for publication, now.